# Emergence of Macrolide-Resistant *Mycoplasma pneumoniae* during an Outbreak in a Primary School: Clinical Characterization of Hospitalized Children

**DOI:** 10.3390/pathogens10030328

**Published:** 2021-03-10

**Authors:** Daniel Hubert, Roger Dumke, Stefan Weichert, Sybille Welker, Tobias Tenenbaum, Horst Schroten

**Affiliations:** 1Pediatric Infectious Diseases, University Children’s Hospital Mannheim, Heidelberg University, 68167 Mannheim, Germany; daniel.hubert@gmx.de (D.H.); Stefan.Weichert@umm.de (S.W.); Horst.Schroten@umm.de (H.S.); 2TU Dresden, Institute of Medical Microbiology and Hygiene, 01307 Dresden, Germany; roger.dumke@tu-dresden.de; 3Institute for Medical Microbiology and Hygiene, Medical Faculty Mannheim, Heidelberg University, 68167 Mannheim, Germany; Sybille.Welker@umm.de

**Keywords:** *Mycoplasma pneumoniae*, outbreak, children, macrolide resistance

## Abstract

*Mycoplasma pneumoniae* (*M. pneumoniae*) is a common causative pathogen of community-acquired pneumonia. Here, we report the development of macrolide resistance during a school outbreak of severe *M. pneumoniae* infections in southwest Germany. We conducted a case series to assess the clinical and laboratory characteristics of hospitalized children with *M. pneumonia* infection and the prevalence of macrolide-resistant *M. pneumoniae* (MRMP) in this patient group. We retrospectively analyzed 23 children with serologically (19 patients) and/or PCR (eight patients) confirmed *M. pneumoniae* infection between October 2019 and December 2019. Most of the 15 hospitalized patients had lower respiratory tract infection (n = 10) and required oxygen therapy (83%). The median length of hospitalization was 7 days (range 3–10 days). In 8/15 patients (53.3%) azithromycin and in 4/15 (26.6%) clarithromycin treatment was applied. However, among the five patients for which extended molecular characterization was performed, sequencing of 23S rRNA revealed no mutation only in the first case, but development of macrolide resistance A2058G in four subsequent cases. Hence, we identified a cluster of hospitalized patients with emerging MRMP. Further studies are warranted to confirm a potential link between macrolide resistance and disease severity.

## 1. Introduction

*Mycoplasma pneumoniae* (*M. pneumoniae*) is one of the most common pathogens causing community-acquired pneumonia (CAP) worldwide, responsible for up to 40% of cases in the general population during epidemic periods. Severe infections occur in children above 5 years of age, whereas younger children often present with a milder clinical course [1,2].

For treatment, tetracyclines and quinolones are not primarily recommended in children due to potential side effects, leaving macrolides as first therapeutic option [1,2]. The prevalence of macrolide-resistant *M. pneumoniae* (MRMP) varies from low levels in Europe (Germany: 3%) to high rates in Asia (China: up to 100%) [3,4].

## 2. Results

During the observational period, overall, 23 children presented either serologically (19 patients) and/or PCR (eight patients) confirmed *M. pneumoniae* infections. Of these 23 children, 15 were hospitalized. Patients (12/15) suffering from pulmonary symptoms were subsequently further analyzed (Table 1). Specific IgM was detected in 9/9 serologically tested patients, PCR was positive in 6/8 patients, and positive serology and PCR were found in 2/5 simultaneously tested patients (Table 2).

Of the 12 patients suffering from pulmonary symptoms, 10 were female and two were male. The median age was 8 years (range 3–10 years). Ten cases (83%) needed oxygen, one additionally required a high-flow nasal cannula. In these 10 cases with clinical signs of pneumonia (83%), chest X-rays were performed and radiological signs of pneumonia were detected in eight cases (80%). Two patients (20%) with oxygen demand had only clinical signs of pneumonia (tachypnea, crackles). The median period of oxygen supplementation was 6 days (range 3–9 days) and the median period of hospitalization was 7 days (range 4–10 days).

Patients with pulmonary symptoms were either treated with azithromycin in eight cases (66%) or with clarithromycin in four cases (33%). A positive test result for *M. pneumoniae* led to a change of antibiotic therapy in four patients, who had been initially treated with β-lactam antibiotics. The first patients with the confirmed wild-type *M. pneumoniae* strain were treated initially with azithromycin.

Overall, seven of eight PCR-positive samples were retrospectively sent to the national reference center for further analysis. In two specimens, PCR amplifications were negative; the remaining five underwent extended molecular characterization. The strains were identical concerning all described typing approaches (P1-, VNTR-, MLST- and SNP-type), suggesting that the outbreak was monoclonal (Table 3). Infection of the first inpatient from our observed cluster proved to be due to a macrolide-sensitive wild-type *M. pneumoniae* strain according to the 23S rRNA sequencing analysis. This changed in the following patients, where an identical nucleotide exchange associated with resistance (A2058G; *Escherichia coli* (*E. coli)* numbering) could be identified. 

No specific outbreak control measures were applied, since case incidence decreased spontaneously. Local health authorities and local outpatient pediatricians were asked to be alert to identify further MRMP infections.

## 3. Discussion

The *M. pneumoniae* strain found in the first investigated patient was sensitive to macrolides, whereas samples taken later during the outbreak from subsequent patients revealed macrolide resistance. To our knowledge, this is the first report about a macrolide-resistant strain emerging during an outbreak in children. Development of resistance during antimicrobial treatment with macrolides has been reported for single patients [5]. Recently, a child with acute lymphoblastic leukemia and chronic *M. pneumoniae* carriage was reported to have a de novo macrolide resistance mutation (mix of A2058T and A2058G) and subsequent breakthrough pneumonia [6]. 

It has been previously discussed that MRMP may cause a more severe clinical course compared to macrolide-susceptible strains [7]. The pneumonia cases in our analysis were hospitalized longer than other cases we usually see in the respective age groups. All sequenced patients were previously healthy and had no macrolide treatment before. Therefore, we suspect either an increased virulence of the strains or that the inadequate treatment with macrolides led to delayed clinical improvement. Moreover, no extraordinary extended infiltrates nor pleural effusion could be detected in the chest X-rays. In a recent large school outbreak, an attack rate of 4% and an extended length of stay of patients in hospital for up to 4 weeks (median 11 days) were observed [8]. Pathogen characterization indicated that *M. pneumoniae* P1 type 1 was the causative agent during this outbreak and the strain harbored an exchange of A to G at position 2058, the most common point mutation in domain V of the 23S rRNA. A recent meta-study showed that the duration of fever in patients infected with macrolide-resistant strains lasts 1.71 days longer and the length of stay is 1.61 days longer [9]. To date, no specific factor has been identified to predict the clinical course of disease, and no evidence supports the causal relationship between macrolide resistance and disease severity.

Due to the retrospective nature of the study, we neither could identify other potential “first” MRMP cases in the outpatient community nor track how many patients in the nearby outpatient community were treated with macrolides. Since the first patient with MRMP in our study cohort was found in October, other potential *M. pneumoniae* infections, either with wild-type or MRMP strains, may have already emerged in the community in September. However, unfortunately, due to a lack of active surveillance and patient material, potential macrolide resistance in these patients could not be further analyzed.

Finally, the necessity and efficacy of antibiotic therapy in *M. pneumoniae* infections is still under debate [9]. All our patients showed a clinical improvement despite proven macrolide resistance, indicating the self-limiting course of the disease. The potential anti-inflammatory effect of macrolides as well as of steroids may contribute to clinical improvement in infections with MRMP [10].

## 4. Materials and Methods

### 4.1. Study Design and Participants

Between October 2019 and December 2019, we observed a cluster of hospitalized pediatric patients with *M. pneumoniae* infections, who went to same primary school in Mannheim. Children with *M. pneumoniae* infection were identified either by serology (IgM/IgG; Institute Virion/Serion, Würzburg, Germany) and/or real-time PCR (FilmArray^®^, Biofire^®^, Nürtingen, Germany) for *M. pneumoniae* (nasopharyngeal swabs). In children for whom patient material was initially available, we analyzed for the presence of macrolide resistance via sequencing of 23S rRNA (German reference center for mycoplasma, Dresden, Germany). 

### 4.2. Data Analysis

Clinical and laboratory characteristics of hospitalized patients with *M. pneumonia* infections were retrospectively analyzed. Due to the small case numbers, a statistical analysis could not be conducted. Instead, a detailed case series description has been performed.

## 5. Conclusions

Our study highlights the probable development of antimicrobial resistance of *M. pneumoniae* during an outbreak of severe cases of CAP among school children. Even in countries with a low rate of resistance, clinicians should be aware of the emergence of resistance of primarily susceptible strains and its potential contribution to the course of subsequent infections. Timely molecular characterization of MRMP strains can help to improve the efficiency of treatment.

## Figures and Tables

**Table 1 pathogens-10-00328-t001:** Clinical characteristics of *M. pneumoniae* infections.

Case	Day of Admission	Age (months)	Gender	Symptoms Before Admission in Days	Temperature in °C	Imaging	Imaging Result	Oxygen Required in Days	Hospitalization Duration in Days	Antibiotic Treatment Before Admission in Days	If Yes, Which Antibiotics?	Antibiotic 1 in Hospital	Antibiotic 2 in Hospital	Discharge Diagnosis	Comments
1	11.10.2019	94	f	10	36.4	chest X-ray	pneumonia	9	10	no		Azithromycin	no	pneumonia	
2	17.10.2019	88	f	9	40.3	chest X-ray	pneumonia	3	8	no		Amoxicillin/Sulbactam	Azithromycin ^1^	pneumonia	Minimal pleural effusion
3	01.11.2019	98	f	8	39.2	chest X-ray	pneumonia	6	8	4	Amoxicillin	Amoxicillin	Clarithromycin ^1^	pneumonia	
4	02.11.2019	100	f	5	37.1	chest X-ray	pneumonia	5	6	no		Clarithromycin	no	pneumonia	
5	03.11.2019	125	f	9	37.3	chest X-ray	pneumonia	6	7	2	Azithromycin	Azithromycin	no	pneumonia	
6	04.11.2019	104	f	6	39.5	chest X-ray	pneumonia	7	8	no		Ampicillin	Clarithromycin ^1^	pneumonia	
7	08.11.2019	55	m	1	36.3	chest X-ray	no infiltrates	2	6	no		Amoxicillin/Sulbactam	Azithromycin ^1^	pneumonia	Ventriculo-peritoneal-shunt
8	12.11.2019	38	f	7	37.2	none	-	n	3	1	Erythromycin	Azithromycin	no	pneumonia	Sibling had pneumonia
9	14.11.2019	100	f	1	38.2	chest X-ray	pneumonia	3	7	no		Azithromycin	no	pneumonia	
10	15.11.2019	143	m	14	36.7	MRI	normal	n	3	no		no	no	neuritis	
11	16.11.2019	105	f	8	37.2	chest X-ray	pneumonia	6	7	3	Cefuroxim	Clarithromycin	no	pneumonia	
12	18.11.2019	115	f	7	38.5	chest X-ray	no infiltrates	6	7	3	Azithromycin	no	no	asthma exacerbation	
13	26.11.2019	57	m	>28	36.4	ultrasound	joint effusion	n	4	no		Azithromycin	no	polyarthritis	
14	30.11.2019	63	m	21	39.8	none	-	n	4	no		Ampicillin	no	pneumonia	
15	13.12.2019	16	m	1	36.4	none	-	n	7	no		no	no	thrombocytopenia	platelets 8.000 (10*9/l)

^1^ Antibiotic treatment was switched from 1st line treatment to 2nd line antibiotic treatment after positive *M. pneumoniae* result was received.

**Table 2 pathogens-10-00328-t002:** Laboratory characteristics of *M. pneumoniae* infections.

Case	Leucocytes (10*9/L)	Neutrophils (%)	CRP mg/L (max)	*M. pneumonia* IgM	*M. pneumonia* IgA	*M. pneumonia* IgG	*M. pneumonia* PCR
1	8.180	n.d.	28.9	n.d.	n.d.	n.d.	pos ^1^
2	10.350	81.2	22.7	pos	neg	neg	pos ^1^
3	10.220	76.3	18.6	pos	neg	20	pos ^2^
4	7.710	56.7	5.1	pos	pos	102	neg
5	5.530	n.d.	15.9	pos	pos	34	n.d.
6	7.540	n.d.	60.5	pos	neg	neg	neg
7	10.270	53.6	71.8	pos	neg	26	n.d.
8	15.620	64.4	49.2	neg	neg	neg	pos ^1^
9	16.140	78.8	29.1	pos	neg	neg	n.d.
10	5.220	n.d.	<2.90	pos	neg	49	n.d.
11	9.730	73.4	39.3	n.d.	n.d.	n.d.	pos ^1^
12	5.420	71.5	16	n.d.	n.d.	n.d.	pos ^1^
13	8.000	66.5	22	pos	neg	44	n.d.
14	17.550	n.d.	51	pos	pos	94	n.d.
15	17.760	48.2	7.8	pos	neg	neg	n.d.

^1^ In these cases, 23S rRNA sequencing analysis was performed. ^2^ Not enough PCR product available for further sequencing. Abbreviations: n.d. = not determined, pos = positive, neg = negative, CRP = C-reactive protein.

**Table 3 pathogens-10-00328-t003:** Results of molecular characterization of *M. pneumoniae* strains.

Case	Age (months)	Macrolide Resistance	P1-Type	VNTR-Typing				MLST-Typing									SNP-Typing								
				13	14	15	16	ppa	pgm	gyrB	gmk	glyA	atpA	arcC	adk	ST ^1^	gyrA	glpK	rpoB	rplB	gmk	442	582	P1	SNP-Type
1	94	Wild-type	1	4	5	7	2	1	2	1	1	1	3	1	1	3	A	A	G	C	A	A	A	G	1
2	88	A2058G ^2^	1	4	5	7	2	1	2	1	1	1	3	1	1	3	A	A	G	C	A	A	A	G	1
8	38	A2058G	1	4	5	7	2	1	2	1	1	1	3	1	1	3	A	A	G	C	A	A	A	G	1
11	105	A2058G	1	4	5	7	2	1	2	1	1	1	3	1	1	3	A	A	G	C	A	A	A	G	1
12	115	A2058G	1	4	5	7	2	1	2	1	1	1	3	1	1	3	A	A	G	C	A	A	A	G	1

1 Sequence type; 2 Overlap A/G. Abbreviations: VNTR = variable number of tandem repeats; MLST = multilocus sequence typing; SNP = single nucleotide polymorphism.

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
