# Peer review of "Emergence of Macrolide-Resistant Mycoplasma pneumoniae during an Outbreak in a Primary School: Clinical Characterization of Hospitalized Children"

_pathogens, 2021, doi:10.3390/pathogens10030328_

Round 1
Reviewer 1 Report
Nothing particular
General:
Huber D et al. concisely described in this paper the five consecutive patients with M. pneumoniae pneumonia in which macrolide-resistance was detected in the four later patient samples. Although the finding may not be very novel for the readers in areas where macrolide-resistant strains are prevalent, otherwise this reviewer has found no major problems to be pointed out in this manuscript.
Specific:
- Abstract, L 23-24. ‘sequencing of 23S rRNA revealed [macrolide susceptibility] only in the first case’; the meaning of [susceptibility] is obscure. ‘Among the five patients for which extended molecular characterization was performed, sequencing of 23S rRNA revealed [no mutation] only in the first case’ might be better.
- Was the first patient treated by either of the macrolides, azithromycin or clarithromycin? Please specify this point.
- How many and which of the five patients, if present, needed oxygen supplementation?
Author Response
Dear reviewer,
We like to thank you, the editorial board and the reviewers very much for the valuable feedback and the helpful comments on our manuscript. In the following, you will find our specific replies to the various comments raised by the reviewers (highlighted with red marking).
Please find attached our revised manuscript. We thank you again for the fruitful suggestions and hope to have met your expectations with the revised version.
Best wishes,
Tobias Tenenbaum
Abstract, L 23-24. ‘sequencing of 23S rRNA revealed [macrolide susceptibility] only in the first case’; the meaning of [susceptibility] is obscure. ‘Among the five patients for which extended molecular characterization was performed, sequencing of 23S rRNA revealed [no mutation] only in the first case’ might be better.
Response: We thank the reviewer for the comment and changed the text accordingly.
Was the first patient treated by either of the macrolides, azithromycin or clarithromycin? Please specify this point.
Response: Yes, the patient received Azithromycin therapy after the positive M. pneumoniae result was optained. The information was included into the manuscript.
How many and which of the five patients, if present, needed oxygen supplementation?
Response: 10 cases needed oxygen therapy. This information was already provided in the text (page 2, line 55)
Reviewer 2 Report
This paper is a brief report of 23 children diagnosed with Mycoplasma pneumoniae (MP) pneumonia during October-December, 2019 in Germany. Macrolide-resistant MP (MRMP) pneumonia rates vary by countries and previous reports in Germany were low by 3%. However among 5 patients which received 23S rRNA sequencing, macrolide resistance was found in 4 cases. Although this is a report of a small cluster in a primary school, this report signifies the possibility of increase in the community. The reviewer has some comments as following.
- Among 12 patients suffering from pulmonary symptoms, 10 cases needed oxygen and 8 cases showed signs of pneumonia. In the 2 cases which did not show signs of pneumonia, what were the reasons for need of oxygen supplementation in the absence of pneumonia?
- The patients admitted in this study, showed high proportion of need of oxygen support. This seems substantially high for MP pneumonia. What are the authors speculations on the reason for oxygen requirement in these children? Were the extent of pneumonia large or was there cases with significant effusion? Did the authors experience non-responsiveness to macrolides as a reason for severity?
- Among patients included in the study, please specify of any underlying diseases or previous history of MP pneumonia.
- The authors state the first inpatient was macrolide-susceptible, and following patients were macrolide-resistant. Were there patients exposed to children treated with macrolides? Could the authors specify or include further epidemiologic data on the outbreak in the school?
Author Response
Dear reviewer,
We like to thank you, the editorial board and the reviewers very much for the valuable feedback and the helpful comments on our manuscript. In the following, you will find our specific replies to the various comments raised by the reviewers (highlighted with red marking).
Please find attached our revised manuscript. We thank you again for the fruitful suggestions and hope to have met your expectations with the revised version.
Best wishes,
Tobias Tenenbaum
Among 12 patients suffering from pulmonary symptoms, 10 cases needed oxygen and 8 cases showed signs of pneumonia. In the 2 cases which did not show signs of pneumonia, what were the reasons for need of oxygen supplementation in the absence of pneumonia?
Response: The 8 case were radiologically confirmed pneumonias, the 2 other cases had tachypnea and crackles, but the chest X-ray was indeterminate. We included the information into the text.
The patients admitted in this study, showed high proportion of need of oxygen support. This seems substantially high for MP pneumonia. What are the authors speculations on the reason for oxygen requirement in these children? Were the extent of pneumonia large or was there cases with significant effusion? Did the authors experience non-responsiveness to macrolides as a reason for severity?
Response: The pneumonia cases were hospitalized longer than other cases we usually see in the respective age groups. Therfore, we suspect either an increased virulence of the strains or the inadequate treatment with macrolides led to delayed clinical improvement. Radiologically, no extraordinary extened infiltrates nor pleural effusion could be detected. We included these aspects into the discussion section of the text.
Among patients included in the study, please specify of any underlying diseases or previous history of MP pneumonia.
Response: All sequenced patients were previously healthy and had no macrolide treatment before. We included these aspects into the discussion section of the text.
The authors state the first inpatient was macrolide-susceptible, and following patients were macrolide-resistant. Were there patients exposed to children treated with macrolides? Could the authors specify or include further epidemiologic data on the outbreak in the school?
Response: To our knowledge, the patients were not exposed to children treated with macrolides. But due to the retrospective nature of the study we have no exact epidemiological data and could not track this information.
Reviewer 3 Report
This case series investigated the development of macrolide resistance in 23 children from October 2019 to December 2019 during a school outbreak of severe Mycoplasma pneumoniae (Mp) infections in Germany. This excellent report identified a cluster of hospitalized patients with emergence of macrolide-resistant Mp (MRMp).
Comments:
- Page 2, line 44: Please present clinical and laboratory characteristics of the 23 patients in a supplementary table (with more detailed information than presented in the paragraph on page 4, line 48-57).
- Page 2, line 45-47: Please show also detailed serological results of the 9 tested patients in a supplementary table (incl. IgM, IgG (IgA?), antibody levels for each isotype, and/or results from paired/convalescent sera if available).
- Page 4, line 62-66: Please illustrate in a figure as timeline graph the investigations (PCR, serology, molecular characterization, MRMp detection), onset of symptoms, hospitalization, antibiotic treatment against Mp, etc. for all 23 patients (as example, please see figure 1, DOI: 10.1093/cid/cir769).
- Page 4, line 76-77: “To our knowledge, this is the first report about a macrolide resistant strain emerging during an outbreak in children.” Please give additional arguments why this MRMp strain should have been emerged during this outbreak and did not develop independently in those children with MRMp detection. Will this be obvious in the figure (see comment above) with subsequent appearance? Were less severely ill children treated ambulatory with macrolides that could have led to the emergence of MRMp and subsequent transmission to those children that later developed more severe disease due to MRMp?
Author Response
Dear reviewer,
We like to thank you, the editorial board and the reviewers very much for the valuable feedback and the helpful comments on our manuscript. In the following, you will find our specific replies to the various comments raised by the reviewers (highlighted with red marking).
Please find attached our revised manuscript. We thank you again for the fruitful suggestions and hope to have met your expectations with the revised version.
Best wishes,
Tobias Tenenbaum
Page 2, line 44: Please present clinical and laboratory characteristics of the 23 patients in a supplementary table (with more detailed information than presented in the paragraph on page 4, line 48-57).
Response: As recommended by the reviewer we included a new figure (now table 1 and 2) including the clinical and laboratory characteristics of the analyzed patients.
- Page 2, line 45-47: Please show also detailed serological results of the 9 tested patients in a supplementary table (incl. IgM, IgG (IgA?), antibody levels for each isotype, and/or results from paired/convalescent sera if available).
Response: Please see comment above. In the new table 2 we also included the Mycoplasma results.
- Page 4, line 62-66: Please illustrate in a figure as timeline graph the investigations (PCR, serology, molecular characterization, MRMp detection), onset of symptoms, hospitalization, antibiotic treatment against Mp, etc. for all 23 patients (as example, please see figure 1, DOI: 10.1093/cid/cir769).
Response: These data were also integrated in the new table 1. Since we don`t have the follow up data of the cases that were treated on an out-patient basis we included only the hospitalized ones in our analysis.
- Page 4, line 76-77: “To our knowledge, this is the first report about a macrolide resistant strain emerging during an outbreak in children.” Please give additional arguments why this MRMp strain should have been emerged during this outbreak and did not develop independently in those children with MRMp detection. Will this be obvious in the figure (see comment above) with subsequent appearance? Were less severely ill children treated ambulatory with macrolides that could have led to the emergence of MRMp and subsequent transmission to those children that later developed more severe disease due to MRMp?
Response: The clinical course of the analyzed patients with M. pneumoniae infection (e.g. longer length of stay) led us to the suspicion that macrolide resistance my play a role here and therefore initiated the molecular testing. The question of the reviewer where and when the macrolide resistance emerged is very interesting. But due to the retrospective nature of the study, we neither could identify other potential “first” MRMP cases in the out-patient community nor track how many patients in the nearby out-patient community were treated with macrolides. Since the first patient with MRMP in our study cohort was found in October, potential other M. pneumoniae infections, either with wild-type or MRMP strains, may have already emerged in the community in September. But unfortunately due to lack of active surveil-lance and patient material potential macrolide resistance in these patients could not be further analyzed. These aspects were included into the discussion.